Health status of Polychrus gutturosus based on physical examination, hematology and biochemistry parameters in Costa Rica

http://orcid.org/0000-0001-5227-6314 Arguedas Randall 1 2 ranarg@gmail.com
Ovares Lizbeth 1
Arguedas Viviana P. 3
Vargas Rodolfo 4
Barquero Marco D. 5
1 FaunaLab , San Jose, San Jose , Costa Rica
2 AWA Science & Conservation , San Jose, San Jose , Costa Rica
3 Recinto de Paraíso, Sede de Atlántico, Universidad de Costa Rica , Montes de Oca, San José , Costa Rica
4 Asociación para el Rescate e Investigación de Vida Silvestre (ASREINVIS), Refugio Animal de Costa Rica , Santa Ana, San José , Costa Rica
5 Sede del Caribe, Universidad de Costa Rica , San Jose, Montes de Oca , Costa Rica
Sotelo-Mundo Rogerio
Electronic publication date: 2021 Jan 6
Publication date: 2021
Volume: 9
Electronic Location ID: e10649
Received 2020 Oct 7; Accepted 2020 Dec 4
Copyright: © 2021 Arguedas et al.
Copyright year: 2021
Copyright holder: Arguedas et al.
License: This is an open access article distributed under the terms of the Creative Commons Attribution License, which permits unrestricted use, distribution, reproduction and adaptation in any medium and for any purpose provided that it is properly attributed. For attribution, the original author(s), title, publication source (PeerJ) and either DOI or URL of the article must be cited.
License URL: https://creativecommons.org/licenses/by/4.0/

Keywords: Hematology, Bush anoles, Costa Rica, Polychrus gutturosus, Clinical pathology, Wildlife health, Lizard, Clinical biochemistry, Arboreal

Funding: Vetlab Costa Rica Escuela de Medicina Veterinaria San Francisco de Asis, Universidad Veritas Vetlab Costa Rica funded all biochemistry analyses. Escuela de Medicina Veterinaria San Francisco de Asis, Universidad Veritas funded the publication fee. The funders had no role in study design, data collection and analysis, decision to publish, or preparation of the manuscript.

==============================
Studies evaluating the health status and characteristics of free-ranging wildlife populations are scarce or absent for most species. Saurian health assessments are usually performed in species that have conservation issues or that are kept in captivity. The Berthold’s bush anole (Polychrus guturossus) is one of eight species belonging to the genus Polychrus, the only representative of the family Polychrotidae. Only a handful of studies have been reported concerning these lizard’s morphological variation, ecology, and natural history, probably because P. gutturosus is a canopy dweller and it can be difficult to locate individuals. It is believed that deforestation and habitat modification could pose a threat for this species, although to date no health assessment has been done. The aim of this study was to generate health baseline data on P. gutturosus. Forty Berthold’s bush anoles (20 males and 20 females) were sampled at the Pacific versant in Costa Rica, where physical examination, skin and cloacal temperatures, and blood samples were obtained from individuals immediately after capture. Animals from the studied population were all healthy (body condition 2.5–3.0/5.0). No lesions or ectoparasites were detected, but hemoparasites were found in nine individuals. Hematological and biochemical values were obtained, and the morphology of leukocytes were found to be similar to other iguanians. A positive correlation was found between the tissue enzymes aspartate aminotransferase (AST) and creatinine kinase (CK) and a negative correlation was found between skin and cloacal temperatures and AST and CK. There were positive correlations between female weight and total protein, calcium, and the calcium and phosphorus ratio. No significant inter-sex differences were found in biochemical values, despite females being larger than males. This is the first health assessment performed on a free-ranging canopy dwelling lizard. These findings provide baseline data that may be useful for future monitoring if the species faces changes in health status due to anthropogenic causes or natural disturbances.

Introduction

Population declines due to anthropogenic causes such as habitat fragmentation, pollution, invasive species, and global climate change are widespread (Sinervo et al., 2010; Brusch, Taylor & Whitfield, 2016). One way to understand how wild animals are impacted by and respond to these environmental stressors is through health assessments (Altizer et al., 2013). Hence, the quantification of hematological and biochemical parameters can be a valuable tool for assessing and monitoring the health and resilience of wild populations (Stacy, Alleman & Sayler, 2011; Campbell, 2014; Maceda-Veiga et al., 2015).

Health assessments are useful when baseline data on normal health parameter values from a clinically robust population are available (Valle et al., 2018). Therefore, it is important to assess the health of wild species, especially populations that have never been surveyed (Valle et al., 2018). This information helps to identify potential effects of disease, injury, pollutants, or other changing environmental conditions that would be difficult to understand without knowledge of normal species-specific variations in hematological and biochemical variables (Smyth et al., 2014; Lewbart et al., 2015). Performing health evaluations on wildlife populations is being utilized more commonly by conservationists (Mathews et al., 2006) and has become a proactive management approach that allows further conservation actions to be taken (Madliger et al., 2017). For example, Henen, Hofmeyr & Baard (2013) found that confiscated adult tortoises showed poorer body condition and lower hematological values than wild ones, while Mathews et al. (2006) found that water voles (Arvicola terrestris) with better body condition and higher hematological values had greater survival probability when reintroduced into the wild.

Studies evaluating the health status and characteristics of free-ranging populations are, however, scarce or absent for most species, especially those that are rarely seen in the wild (Bell & Donnelly, 2006; Whitfield et al., 2007; Dallwig et al., 2011). In lizards, health assessments reported in the literature have usually been done on species that are threatened (Alberts et al., 1998; Espinosa-Avilés, Salomón-Soto & Morales-Martínez, 2008; McEntire et al., 2018), endemic (Lewbart et al., 2015; Arguedas et al., 2018), or kept in captivity (Ellman, 1997; Mayer et al., 2005; Laube et al., 2016), providing information on the survival of species with conservation issues. However, free-ranging species with no apparent threats have generally not been evaluated as well.

The Berthold’s bush anole (Polychrus guturossus) is one of eight species belonging to the genus Polychrus and the only representative of the family Polychrotidae in Middle America. This is a moderately large, diurnal lizard that is distinguished by its bright green body coloration and extremely long tail (over three times the length of the head and body) (Savage, 2002). The species is sexually dimorphic, with females being larger than males (Savage, 2002; Koch et al., 2011) and females having green eyelids, while males have yellow eyelids. The species ranges from Honduras to northwestern Ecuador, apparently restricted to moist and wet forests (Savage, 2002; Leenders, 2019). Despite its large distribution, only a handful studies have been carried out concerning its morphological variation, ecology, and natural history (Taylor, 1956; Roberts, 1997; Koch et al., 2011; Gómez-Hoyos et al., 2015; Bringsøe, Alfaro Sánchez & Hansen, 2016; Ruiz, Gutiérrez & Flóres Rocha, 2016). Polychrus gutturosus is a canopy dweller and its body coloration makes it difficult to locate individuals during daylight hours. It is believed that deforestation and habitat modification could pose a threat for this species (Acosta Chaves et al., 2017), although no health assessment has ever been done and population status is unknown.

Health assessments of wildlife in Costa Rica are rare. To our knowledge, health evaluations of free-ranging species have been performed on 20 mammals (Schinnerl et al., 2011; Hagnauer Barrantes, 2012; Bernal-Valle, Jiménez-Soto & Meneses-Guevara, 2020) and only one reptile (green basilisk, Basiliscus plumifrons, Dallwig et al., 2011). Therefore, our aim is to generate data to improve our knowledge of the health status of more Costa Rican reptiles, by providing baseline data on a wild population of the unique lizard species (P. gutturosus). The following baseline data was included: (1) body temperature and weight, (2) presence of ectoparasites and external abnormalities through physical examination, and (3) hematological and biochemical values. Most of our knowledge on P. gutturosus comes from museum specimens (Savage, 2002; Koch et al., 2011) and sporadic observations of individuals in the field (Gómez-Hoyos et al., 2015; Bringsøe, Alfaro Sánchez & Hansen, 2016; Ruiz, Gutiérrez & Flóres Rocha, 2016). Therefore, this is the first long-term, empirical study on free-ranging P. gutturosus and one of the few studies overall that has been carried out on a species inhabiting the forest canopy. Our data was also compared to similar information previously published for close relatives of P. gutturosus.

Materials and Methods

Ethics statement

All research methods were authorized by Costa Rica’s National System of Conservation Areas (SINAC) under permit numbers SINAC-ACC-PI-R-102-2018 and SINAC-ACC-257-2018.

Animal collection and handling

A total of 40 adult individuals (20 males and 20 females) were collected from October 2018 to May 2019, carrying out one field trip per month. Lizards were surveyed along a public, dirt road at El Rodeo (Cascante-Marín, 2012), Ciudad Colón, San José, Costa Rica (Fig. 1). The area has an irregular topography ranging from 400 to 1016 meters above sea level (masl) and with an annual average temperature of 23.4 °C and an annual average rainfall of 2,467 mm (Cascante-Marín, 2012). Two seasons are evident, a rainy season from May to October and a dry season from December to March, with two transitional months (November and April) (Cascante-Marín, 2012). The area of El Rodeo shows a landscape composed of pastureland and agricultural and urban zones (Fig. 1), although the road sampled was surrounded by bushes, shrubs and trees on both sides. The lizards were searched for only on such shrubs and trees at night, since resting animals are easier to spot. Animals were located between 560 and 754 masl and air temperatures ranged from 21.2 °C to 27.7 °C.

Figure 1 Map of Costa Rica showing the location of the Berthold’s bush anole (Polychrus gutturosus) sampling site with exact coordinate points along an approximately 3 km trail showing where they were captured.

Map of Costa Rica showing the location of the Berthold’s bush anole (Polychrus gutturosus) sampling site with exact coordinate points along an approximately 3 km trail showing where they were captured. The yellow marks refer to collection points, not to individuals.

Once an individual was observed, the skin temperature at the resting site was measured using a digital laser infrared thermometer gun (Nubee®, NUB8550AT model). The lizards were then hand-caught from shrubs or trees and taken to a workstation at the temporary mobile field laboratory approximately 5 m to 10 m from the collection site. A J/K/T/E thermocouple thermometer (1312 model; Professional Instruments®, Hopkins, MN, USA) was used to measure the lizard’s cloacal body temperature, which was taken by inserting the K probe into the cloaca, approximately 1–2 min after capture. A blood sample was taken after the temperature was measured. The process from catching the animal to collecting its temperature and blood lasted about 12 min.

Physical examination and tagging

The individual was placed in a cloth bag and weighed on a digital scale (to the nearest 0.1 g). Afterwards, the lizard was examined for obvious abnormalities or lesions. Physical examinations were performed according to Divers (2019). Oral cavity inspection was easily performed since they kept their mouths open as a defense mechanism. Any external parasites found on the skin were noted and females were gently palpated to detect if they were gravid (feeling for palpable eggs). The body condition was assessed on a scale of 1–5; 1 being emaciated, 2 underweight, 3 normal, 4 overweight, 5 obese (Divers, 2019). After physical examination, a blood sample was collected and each anole was measured to determine snout-vent length (SVL), and then tagged subcutaneously in the left inguinal region (https://wsava.org/global-guidelines/microchip-identification-guidelines/) with a Biomark® HPT12 radio frequency identification tag and released back where it was collected.

Hematology and biochemistry analyses

Each lizard was manually restrained and 0.2–0.4 ml of blood was drawn from the ventral coccygeal vein. If two attempts to collect blood from the tail were unsuccessful, then blood was taken from the jugular vein. Blood sampling time varied between 3 and 5 min. For blood draws, a heparinized 30-gauge needle attached to a 1.0 ml syringe was used. Two blood films were immediately made on clean glass microscope slides and then the rest of the sample from the syringe was placed in a 0.5 ml Eppendorf® tube. All samples were taken to the laboratory the same night and stored at 4 °C to be processed the following day. Red blood cell (RBC), white blood cell (WBC) and thrombocyte count (TC) were performed using the standard method of a Natt and Herrick solution (1/200) on a Boeco® Neubauer Improved chamber. Packed cell volume (PCV) was determined using high-speed centrifugation (Digisystem® Laboratory Instruments Inc., New Taipei City, Taiwan) of blood-filled microhematocrit tubes. Differential white blood cells were obtained by examining a peripheral smear stained with Diff-Quick® stain (Campbell, 2014). Polychromatophil percentage was determined by counting the number of polychromatophils among 1,000 erythrocytes.

Total proteins were obtained by means of a clinical refractometer (REC-200ATC®, RETK-70 model) using plasma from the microhematocrit tube. Biochemical parameters such as aspartate aminotransferase (AST), albumin (Alb), calcium (Ca), cholesterol (Chol), creatinine kinase (CK), glucose, phosphorus (P) and uric acid (UA) were measured with a Roche® analyzer (Cobas c111 model) following the company’s instructions.

Comparison with close relatives

Literature was reviewed for similar hematology and biochemistry information published on close relatives of the Berthold’s bush anole. The review focused on the infraorder Iguania, which includes P. gutturosus, according to the phylogeny proposed by Pyron, Burbrink & Wiens (2013). Twenty-nine articles were found (see Supplemental Data) corresponding to eight of the 14 families that make up Iguania, from which the mean and standard deviation (or range, when SD was not reported) of hematological and biochemical parameters of free-ranging individuals was obtained. This information was used to place the physiological values generated for P. gutturosus within a phylogenetic context.

Statistical analyses

The mean, standard deviation, range, and 95% confidence intervals for all blood parameters were calculated. Differences in weight, biochemistry and hematological values between the sexes were examined using t-tests. Differences between animals infected with hemoparasites and non-infected animals in terms of PCV, RBC, heterophil to lymphocyte (H:L) ratio, WBC, weight and SVL and Scaled Mass Index (SMI) and sexes were determined using t-tests. A Pearson correlation was calculated to look at the association between body temperature (skin and cloacal), weight and SVL with all the hematologic and biochemistry values. To estimate body condition, the SMI was used. This index proved to be a better indicator of the relative size of energy reserves and other body components, SMI = Mi (Lo/Li) bSMA (Peig & Green, 2009). The length (Li) variable has the strongest correlation with mass (Mi) on a log-log scale, since this is likely to be the length that best explains that fraction of mass associated with structural size. The scaling exponent (bSMA) is calculated indirectly by dividing the slope from an ordinary least squares regression and Lo is the mean of the total sample length (Peig & Green, 2009). All statistical analyses were performed using IBM SPSS®v24 with a standard α level of 0.05. In addition, information from nine sample points from 40 Berthold’s bush anoles were geocoded and a map was generated using ArcGis 10.1 software (ESRI, Redlands, CA, USA).

Results

Physical examinations

All lizards appeared to be active and healthy. Female weight ranged from 27 g to 80 g (mean ± SD = 52.25 ± 13.56) and males weighed from 17 g to 52 g (mean ± SD = 37.30 ± 8.86). No evidence of lesions was detected during physical exams. No ectoparasites (acari, ticks, or other macroscopic arthropods) were observed and none of the females had palpable oviductal eggs.

The general body condition of all individuals was between 2.5 and 3.0 and a body mass index was also obtained. The SMI was 3.75 (±0.15) CI [3.70–3.79]. No significant differences were found between sexes (t = 0.99, p = 0.33).

Physiological parameters

No significant differences between sexes in any of the hematological or biochemical parameters were found. Hematological values are presented in Table 1. The morphology of lymphocytes (Fig. 2A), heterophils (Fig. 2B), eosinophils (Fig. 2C), basophils and monocytes (Fig. 2D) were similar to other iguanian species.

Table 1 Hematological values (n = 40) of the Berthold’s bush anole (Polychrus gutturosus) in Costa Rica.

PCV (Packed cell volume), RBC (Red blood cells), and WBC (White blood cells).

Analyte (Units)	Mean ± SD	Range	95% CI	
PCV (%)	31.75 ± 4.53	23.00–44.00	[30.35–33.15]	
RBC (1012/L)	0.94 ± 0.20	0.64–1.35	[0.88–1.01]	
Polychromatophils (%)	1.33 ± 0.69	0.4–3.0	[1.11–1.54]	
WBC (109/L)	19.44 ± 6.66	8.04–37.18	[17.38–21.51]	
Thrombocyte Count (109/L)	2.13 ± 1.14	0.21–4.52	[1.78–2.49]	
Heterophils (109/L)	2.66 ± 1.36	0.84–6.99	[2.23–3.08]	
Heterophils (%)	13.78 ± 5.14	6.00–29.00	[12.18–15.37]	
Lymphocytes (109/L)	14.37 ± 5.36	5.87–27.14	[12.71–16.03]	
Lymphocytes (%)	74.13 ± 10.01	26.00–87.00	[71.02–77.23]	
Monocytes (109/L)	1.76 ± 1.71	0.12–10.13	[1.22–2.29]	
Monocytes (%)	8.60 ± 6.62	1.00–42.00	[6.55–10.65]	
Eosinophils (109/L)	0.58 ± 0.41	0.00–1.57	[0.45–0.70]	
Eosinophils (%)	3.03 ± 1.79	0.00–7.00	[2.47–3.58]	
Basophils (109/L)	0.11 ± 0.15	0.00–0.45	[0.06–0.16]	
Basophils (%)	0.58 ± 0.75	0.00–2.00	[0.34–0.81]	
H:L Ratio	0.21 ± 0.17	0.08–1.12	[0.15–0.26]	

Figure 2 Photographs of selected Berthold’s bush anole (Polychrus gutturosus) blood cells stained with Diff-Quick stain at 100×.

(A) heterophil (B) basophil (C) lymphocyte (D) monocyte (E) intraerythrocytic hemoparasite.

Heterophil to lymphocyte (H:L) ratios were calculated (Table 1). Both shape and appearance of erythrocytes and thrombocytes were similar to those reported for other reptiles. Erythrocytes were ellipsoid with central positioned oval nucleus that contain dense purple chromatin with rather irregular margins. The cytoplasm stained orange o pale pink with Diff-Quick. Thrombocytes are elliptical with the nucleus located in a central position, containing dense chromatin that stained purple. The cytoplasm is colorless. Polychromatophilic erythrocyte percentage was 1.33 (±0.69) (Table 1). Intraerythrocytic parasites were found in nine (three females and six males) of the 40 individuals (22.5% of the total sample) (Fig. 2E). No significant differences were found between individuals with and without hemoparasites for the following variables: PCV (t = −1.24, p = 0.22), RBC, (t = 1.11, p = 0.27), WBC (t = 0.64, p = 0.52), H:L ratio (t = 1.55, p = 0.28), weight (t = −0.16, p = 0.86), SVL (t = −1.43, p = 0.16), and polychromasia (t = −0.64, p = 0.53).

Clinical biochemistry values are reported in Table 2. A wide range was observed in AST (15.1–139.40 U/L) and CK (122.9–6,848.20 U/L), and both muscle enzymes were highly correlated (r = 0.795, p < 0.001). Skin temperature varied between 18.8 °C and 26.2 °C (mean ± SD = 22.31 ± 1.74) and cloacal temperature varied between 21.2 °C and 32.4 °C (mean ± SD = 25.22 ± 2.11). A negative correlation was found between skin temperature (r = −0.51, p = 0.001) and cloacal temperature (r = −0.42, p = 0.007) with AST (Fig. 3A). The same occurred between skin temperature (r = −0.51, p = 0.001) and cloacal temperature (r = −0.42, p = 0.007) with CK (Fig. 3B). A positive correlation was found in females, but not in males, between calcium (r = 0.57, p = 0.009), total protein (r = 0.49, p = 0.03) and the calcium/phosphorus (Ca/P) ratio (r = 0.71, p < 0.001) with weight.

Table 2 Blood biochemical values (n = 40) of the Berthold´s bush anole (Polychrus gutturosus) in Costa Rica.

A/G Ratio (Albumin/Globulin ratio), AST (Aspartate aminotransferase), CK (Creatinin kinase), and Ca:P Ratio (Calcium:Phosphorus ratio).

Analyte (Units)	Mean ± SD	Range	95% IC	
Glucose (mmol/L)	11.96 ± 2.04	8.38–16.10	[11.32–12.59]	
Total Protein (g/L)	75.10 ± 7.80	60.00–90.00	[72.68–77.52]	
Albumin (g/L)	17.91 ± 6.34	3.70–28.17	[15.95–19.88]	
Globulins (g/L)	57.19 ± 6.20	46.10–70.32	[55.27–59.11]	
A/G Ratio	0.32 ± 0.12	0.06–0.52	[0.28–0.36]	
AST (U/L)	35.08 ± 23.86	15.10–139.40	[27.69–42.47]	
CK (U/L)	1,283.56 ± 1,366.22	122.90–6,848.20	[860.17–1,706.94]	
Calcium (mmol/L)	3.81 ± 1.64	2.20–9.35	[4.32–3.30]	
Phosphorus (mmol/L)	2.46 ± 0.85	1.43–5.49	[2.72–2.20]	
Ca:P Ratio	1.60 ± 0.55	0.76–3.47	[1.43–1.77]	
Uric acid (µmol/L)	223.78 ± 209.56	59.30–1,164.90	[158.84–288.72]	
Cholesterol (mmol/L)	8.97 ± 3.16	4.25–17.50	[7.99–9.95]	

Figure 3 Linear correlations of skin and cloacal temperatures (°C) of the Berthold’s bush anole (Polychrus gutturosus) with blood values of (A) aspartate aminotransferase (AST) and (B) creatinine kinase (CK).

Phylogenetic comparison

Even though hematological and biochemical information is not available for a number of iguanian families (e.g., Leiocephalidae, Crotaphytidae, Hoplocercidae, Opluridae and Leiosauridae), some comparisons are still possible. For hematological parameters, WBC was found to be higher for Polychrotidae, Liolaemidae and Corytophanidae (all three phylogenetically related) compared to other families, while the number of lymphocytes is high in Polychrotidae and comparable with Iguanidae and Tropiduridae (Fig. 4). For biochemical parameters, Polychrotidae showed a higher value of total protein when compared to Tropiduridae, although no other value differed significantly (Fig. 5).

Figure 4 Mean ± SD (or range) of hematological parameters extracted from the literature for species of the infraorder Iguania, as a comparison with the Berthold’s bush anole (Polychrus gutturosus).

Also depicted are the phylogenetic relationships (adapted from Pyron, Burbrink & Wiens (2013)) of iguanian families for which hematological information was available. The full name of each species is: Furcifer pardalis (panther chameleon) Laube et al. (2016); Intellagama lesueurii (Australian water dragon) Johnson et al. (2018); Microlophus bivittatus (San Cristóbal lava lizard) Arguedas et al. (2018); Amblyrhynchus cristatus (marine iguana) Lewbart et al. (2015); Cyclura cychlura (Andros Island iguana) James et al. (2006), Phrynosoma cornutum (Texas horned lizard) McEntire et al. (2018); Liolaemus wiegmannii (Wiegmann’s lizard) Ceballos de Bruno (1995) and Basiliscus plumifrons (green basilisk) Dallwig et al. (2011). ND = No Data.

Figure 5 Mean ± SD (or range) of biochemical parameters extracted from the literature for species of the infraorder Iguania, as a comparison with the Berthold’s bush anole (Polychrus gutturosus).

Also depicted are the phylogenetic relationships (adapted from Pyron et al. (2013)) of iguanian families for which biochemical information was available. The full name of each species is: Furcifer pardalis (panther chameleon), Intellagama lesueurii (Australian water dragon) Johnson et al. (2018); Pogona vitticeps (bearded dragon), Microlophus bivittatus (San Cristóbal lava lizard) Arguedas et al. (2018); Iguana iguana (green iguana) Harr et al. (2001); Cyclura cychlura (Andros Island iguana); James et al. (2006) and Basiliscus plumifrons (green basilisk.) Dallwig et al. (2011). ND = No Data.

Discussion

Health assessments provide baseline information that can be used to understand future changes in the health status of wildlife populations. Both physical examination and internal physiological data (i.e., body temperature, hematology and biochemistry) can serve as valuable tools for evaluating and monitoring the health of wild populations (Stacy, Alleman & Sayler, 2011; Campbell, 2014), especially when such assessments provide the only available data for a given species (Innis, 2014). Although physical examinations are common in many taxa, including reptiles, there are no known reports assessing hematology and biochemistry parameters in free-ranging, canopy dwelling lizards. Therefore, this study is important to report such data for P. gutturosus.

Physical examination of Berthold’s bush anoles showed no evident abnormalities, suggesting that all animals were apparently healthy. Body condition is assumed to influence an animal’s health and fitness (Peig & Green, 2009) and although the body condition index cannot be compared with other studies, the fact that no differences were found between sexes indicates an evenness to our sample. These findings may indicate that environmental conditions such as availability of habitat, food and water are fulfilling the requirements of the individuals of the population studied, despite being located in an altered area (Fig. 1). Furthermore, healthy animals also suggest that physiological parameters, such as hematological and biochemical blood values, may be within a normal range. Blood cell counts and cell morphology, however, are highly variable between reptilian species, even among members of the same genus (Stacy, Alleman & Sayler, 2011; Innis, 2014). Such variation is caused by both intrinsic and extrinsic factors like age, sex, season, presence of environmental stressors, parasite load, nutritional status, and capture and restraint (Campbell, 2014; Heatley & Russel, 2019). For that reason, the results in this study are compared with other related lizard species.

Packed cell volume and RBC counts were similar to closely related lizards (James et al., 2006; Dallwig et al., 2011; McEntire et al., 2018), and polychromatophilic cell mean was 1.33%. In normal reptiles, the percentage of polychromatophilic red cells is from >1% to 2.5% (Heatley & Russel, 2019). Erythrocyte counts and the presence of a high percentage of polychromasia have been used as an important parameter for health assessments of wild lizards. For example, Smyth et al. (2014) found that sleepy lizards (Tiliqua rugosa) in agricultural environments had a regenerative anemia (low PCV and increased polychromatophils) compared to animals in non-agricultural areas.

In squamate species, lymphocytes are the predominating circulating cell, usually 80% of the leukogram (Sykes & Klaphake, 2015; Heatley & Russel, 2019), although in some species heterophils can be the main circulating leukocyte. Hematological data comparisons with other closely related families showed that lymphocytes were the main white cell population in P. gutturosus, followed by heterophils and monocytes (Fig. 5). For example, Polychrus, Amblyrhynchus, Microlophus, Intellagama and Furcifer include species that are predominantly lymphocytic, while heterophils are the predominant circulating leukocyte cell in Basiliscus, Cyclura, Phrynosoma and Liolaemus (Fig. 5). These blood circulating cells are important in calculating the heterophil to lymphocyte ratio (H:L ratio) which has been used as an indicator of stress in reptiles (Aguirre et al., 1995; Cartledge, Gartrell & Jones, 2005; Davis, Maney & Maerz, 2008; French, Fokidis & Moore, 2008; Silvestre, 2014) and wild and domestic birds (Vleck et al., 2000; Huff et al., 2005).

Normal H:L ratio in reptile species with more lymphocytes circulating than heterophils will have potentially delayed responses to heterophilia (Davis, Maney & Maerz, 2008; Campbell, 2014; Silvestre, 2014). It has also been demonstrated that in other vertebrates with low H:L ratio (which means that lymphocytes are the predominant cell), H:L increase rapidly after stress events (Cīrule et al., 2012). Although we are aware of H:L limitations in reptiles, we consider that this measurement is important for animal health assessments.

Most biochemistry analytes measured in P. gutturosus were within similar ranges of other iguanian species. CK and AST values were similar to those found in Cyclura species (Alberts et al., 1998; James et al., 2006; Maria et al., 2007) and Basiliscus plumifrons (Dallwig et al., 2011), in which the length of the capture, holding period, restraint and venipuncture results in elevated CK and AST levels. In reptiles, CK is an enzyme considered to be specific to muscle cells and thus with muscle damage will elevate in the blood, while AST is a less specific enzyme and is primarily in liver but also in muscle tissue (Anderson et al., 2013; Bogan & Mitchell, 2014; Petrosky, Knoll & Innis, 2015). A high positive correlation between AST and CK was found in P. gutturosus, suggesting that higher levels of the enzyme AST may be associated with muscle tissue along with CK in this lizard species.

A negative correlation was found between AST and CK and both skin and cloacal temperatures (Fig. 4). As ectotherms, reptiles experience temperature-induced changes in metabolic rate (Niewiarowski & Waldschmidt, 1992). When reptiles are resting and their body temperature is low, their metabolic rate and energy stay at basal levels (Vitt & Caldwell, 2014); however, movement or using anaerobic metabolism in specific situations require more energy than the basal rate, so reptiles attain higher body temperatures (Randall et al., 2002). During high-intensity, short-duration activity (e.g., capture and sampling of the lizards (see “Materials and Methods”)), the concentration of ATP within muscles can be maintained constant by continuous re-phosphorylation of ADP by the CK reaction (Randall et al., 2002). As a result, an animal can use the large reserve of high-energy phosphate in CK to power muscle contraction until oxidative and anaerobic metabolism start to generate ATP, allowing it to move for much longer (Randall et al., 2002). Since our sampling (capture, restrain and venipuncture) was performed at night, individuals of P. gutturosus had lower body temperatures and thus likely lower oxygen consumption (Clark, Butler & Frappell, 2006), such that muscle contraction rapidly started using anaerobic (glycolytic) pathway to keep its activity (Bennett, 1980). Anaerobic muscular metabolism also generates an electrolyte imbalance (mainly calcium) and releases oxygen and lactate, leading to muscle injury (Giannoglou, Chatzizisis & Misirli, 2007). Such muscle damage causes CK and AST enzymes to leak into the blood stream from muscle cells (Allison, 2005). Hence, increased plasma activities of both CK and AST suggest active or recent muscle injury (Silvestre, 2014). Therefore, animals at lower temperatures, with lower oxygen consumption, utilized the anaerobic pathway at the moment of capture, leading to more muscle damage, resulting in the release of more CK and AST than lizards captured at higher temperatures, which probably utilized anaerobic muscular activity later.

No differences between sexes were found among hematological and biochemical variables. In other iguanian lizards where males are larger than females, significative differences in biochemical values have been found between sexes (Dallwig et al., 2011). For example, males of the San Cristóbal lava lizard (Microlophus bivittatus) had higher hemoglobin, PCV and glucose than females (Arguedas et al., 2018), and female green iguanas (Iguana iguana) had higher hemoglobin and PCV than males (Harr et al., 2001). Interestingly, in Phrynosoma cornutum, where females are larger than males, basophil counts were lower in females than in males (McEntire et al., 2018). Most explanations for differences between sexes in hematological and biochemical values are based on reproductive physiological status or hormonal biases (McEntire et al., 2018), although the reasons why P. gutturosus have no difference between sexes are unknown.

A positive correlation between calcium and proteins with body weight was found for females but not for males. It is known that during vitellogenesis, circulating estrogens raise calcium, phosphorus and proteins in plasma (Bonnet, Naulleau & Mauget, 1994; Jones, 2011), however, no correlation between P and weight was found. Calcium increases during vitellogenesis and folliculogenesis for most squamates, the investment of calcium in eggshells is considerably less than for yolk (Stewart & Ecay, 2010). We hypothesize that heavier females may be under active vitellogenesis, increasing their weight due to follicular development.

A correlation was found between calcium to phosphorus (Ca/P) ratio and weight in females but not in males. Ca and P homeostasis are directly interrelated because serum Ca interplays with serum P through the modulation of several hormones, such that serum concentration is approximately inversely related (a high Ca/P ratio means higher Ca than P) (Madeo et al., 2018). Calcium increases proportionally greater than P, resulting in a higher value of Ca/P. Although the reason for that is unknown, a possible explanation is that parathyroid activity may be higher in heavier females due to larger follicular development. Unfortunately, no literature is available regarding the breeding season on this species, so the reproductive stage of the animals sampled is unknown.

Finally, intraerythrocytic parasites were found in nine individuals, but no differences were found between infected and non-infected animals with hematological values or physical measurements. The presence of hemoparasites in wild reptiles is common (Telford, 2009) and usually considered non-pathogenic (Stacy, Alleman & Sayler, 2011). Hemoparasite life cycles involve sexual reproduction in an invertebrate host (e.g., ticks, mites, mosquitoes and flies) and asexual reproduction in the reptilian host (Telford, 2009; Campbell, 2015). Since no mites or ticks were found in the lizards sampled (which may be due to their arboreal habits), it is possible that the hemoparasites were transmitted by mosquitoes or flies. Pathogenesis caused by hemoparasite infections in reptiles is unclear, with studies reporting from apparently non-detrimental infections in natural hosts to severe and life-threatening illness in unnatural hosts (Maia et al., 2014). Hence, more research is needed to identify the species of hemoparasite identified here and continued monitoring of these lizard populations to establish actual prevalence of the disease.

Health assessments allow for evaluation of body condition, disease, stress levels, hydration status and temperature changes of wild populations to be detected (Stacy, Alleman & Sayler, 2011; Innis, 2014) and thus, determine whether a population faces any stress related to environmental changes or anthropogenic causes. In this study, the first baseline data of hematology and clinical biochemistry values for the Berthold’s bush anole (P. gutturosus) is reported. Poorly studied species with populations occurring in altered or non-protected environments can be at greater risk from human activity. Since P. gutturosus depends on its arboreal habits, deforestation due to urban or agricultural activities can affect their survival, reduce its habitat and increase the transmission of diseases. Preventing species from becoming threatened requires conservation actions based on scientific knowledge. This includes health assessments of wild populations that can be used for future management and protective actions.

Conclusions

Hematological and biochemical values were obtained for the first time in this poorly studied arboreal lizard species. The morphology of leukocytes were similar to other iguanians. A positive correlation was found between aspartate aminotransferase (AST) and creatinine kinase (CK) and a negative correlation between skin and cloacal temperatures with AST and CK. There were positive correlations between female weight and total protein, calcium, and the calcium and phosphorus ratio. No significant inter-sex differences were found, despite females being larger than males. These findings provide baseline data that may be useful if this species faces changes in health status due to anthropogenic causes or natural disturbances in the future.

Supplemental Information

Supplemental Information 1 Additional clinical pathology references of lizards.

Saurian hematology and biochemistry for comparison.

Click here for additional data file.

Supplemental Information 2 Raw data.

Click here for additional data file.

We thank Esteban Castro from VetLab for his help on processing the samples and Mario Baldi from the National University of Costa Rica School of Veterinary Medicine, for elaborating the map. Also, we extend our gratitude to Aaron Solís, Edwin Soto and José Gabriel Barquero for helping in the field. Lastly, we are thankful to Belinda Dick for English editing.

Additional Information and Declarations

Competing Interests

Author Contributions

Animal Ethics

Data Availability

The authors declare that they have no competing interests.

Randall Arguedas conceived and designed the experiments, performed the experiments, analyzed the data, prepared figures and/or tables, authored or reviewed drafts of the paper, and approved the final draft.

Lizbeth Ovares conceived and designed the experiments, performed the experiments, prepared figures and/or tables, authored or reviewed drafts of the paper, and approved the final draft.

Viviana P Arguedas conceived and designed the experiments, performed the experiments, prepared figures and/or tables, authored or reviewed drafts of the paper, and approved the final draft.

Rodolfo Vargas conceived and designed the experiments, performed the experiments, authored or reviewed drafts of the paper, and approved the final draft.

Marco D Barquero conceived and designed the experiments, performed the experiments, analyzed the data, prepared figures and/or tables, authored or reviewed drafts of the paper, and approved the final draft.

The following information was supplied relating to ethical approvals (i.e., approving body and any reference numbers):

All research methods were authorized by Costa Rica’s National System of Conservation Areas (SINAC) under permit numbers SINAC-ACC-PI-R-102-2018 and SINAC-ACC-257-2018.

The following information was supplied regarding data availability:

The data are available in the Supplemental Files.

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
