# Peer review of "Health status of Polychrus gutturosus based on physical examination, hematology and biochemistry parameters in Costa Rica"

_PeerJ, doi:10.7717/peerj.10649_

## Round 0.1 · original submission · Minor Revisions

Please take into consideration the reviewer’s comments and provide back a point-by-point rebuttal letter addressing those concerns.

·

Basic reporting

It is a very good paper, with an adequate experimental design wich offers new and valuable biological and physiological information about a non well know lizard specie from Central America.
However, need some corrections or adequations in order to be a paper of excellence
The literature provided is appropriate and sufficient.

Experimental design

It is a good research paper, with a very good experimental design that fall into the scope of the journal

Validity of the findings

The paper offers valuable biological and physiological information about a non well know new lizard species.Statistical analysis is well designed

Additional comments

Materials and methods
Lines 150-151: How much animals were bleed twice ? (Caudal and jugular vein) This procedure can affect the hemogram and some biochemical analytes as a CPK or AST. In this case, the sample was processed or discarded?and in the case that the sample was processed, the results of this duplicated samples were included?
Results Lines 215-216
The erythrocytes and thrombocytes shapes and dimensions were measured ? Because in the text the authors write” Both shape and appearance of erythrocytes and thrombocytes were similar to those reported for other reptiles.” Please, it is necessary to clarify this point with a brief descriptions of erythrocytes and thrombocytes morphology, shape and appearance.

Discussion,Lines 286-292
Please clarify the point about the validity of the ratio Heterophiles: Lymphocytes as a tool to assay the stress, taking in mind that H:L ratio is based in species wich have heterophiles as a the predominant white blood cell. In stress condition a decreased lymphocyte count occurs( lymphopenia) and appears a concomitants heterophilia . In this cases, wich the animals exhibiting linfocytes as a predominant leukocytes, is not clear the adequate use of this ratio as an health status evaluation tool. A lot of studies suggest that H : L ratios can be used to assess glucocorticoid levels and stress in reptiles; however, it is point out that less work has been conducted in the reptilian taxon than in others vertebrates.
Figure 2.c: The cell described as an eosinophile is not this type of leukocyte .The eosinophile have reddish,round and regular sized granules, well visibles with the stain,and the nucleus is eccentrically located. In the picture i can see a round cell, with a unique and central nucleus and a lot of colorless vesicles, without granules. Probably is another type of blood cell , but ,in this case , unfortunately not an eosinophile. Please verify the picture and the stain.

·

Basic reporting

no comment

Experimental design

The submission was clearly defined the research question. The study method was conducted rigorously and to a enough technical standard. Methods was described with sufficient information to be reproducible by another researchers.

Validity of the findings

All underlying data have been provided; they are robust, statistically sound, and controlled. Conclusions are well stated, linked to original research question and limited to supporting results.

Additional comments

Dear Authors,
I have gone through your submission about the hematology and some biochemistry parameters of Polychrus gutturosus in Costa Rica. Unfortunalety, there are limited data on baseline data of healty parameters in many herptile species. The present study will be fill the gap in this subject for Polychrus gutturosus. However, I noticed there are some minor point which need revise or remove. I indicate my suggestion on text. In my opinion, the article should be accepted if the authors make some revisions, minor enough that I would NOT necessarily need to re-review it.

·

Basic reporting

Approved. See specific comments, suggestions, recommendations in PDF review. Specific changes recommended on several figures and tables.

Experimental design

Approved. See comments and general recommendations in PDF review.

Validity of the findings

Approved. See comments in PDF review.

Additional comments

See notes embedded in review of PDF.
Overall well put together study and paper. See comments and suggestions in PDF review.

---

## Round 0.2 · accepted · Accept

Thanks for addressing all the revisions and corrections requested. Now your manuscript is accepted in PeerJ.